# Stimulating Mitochondrial Biogenesis with Deoxyribonucleosides Increases Functional Capacity in ECHS1-Deficient Cells

**DOI:** 10.3390/ijms232012610

**Published:** 2022-10-20

**Authors:** Harrison James Burgin, Jordan James Crameri, Diana Stojanovski, M. Isabel G. Lopez Sanchez, Mark Ziemann, Matthew McKenzie

**Affiliations:** 1School of Life and Environmental Sciences, Faculty of Science, Engineering and Built Environment, Deakin University, Geelong, VIC 3216, Australia; 2Department of Biochemistry and Pharmacology, Bio21 Molecular Science and Biotechnology Institute, The University of Melbourne, Melbourne, VIC 3010, Australia; 3Centre for Eye Research Australia, Royal Victorian Eye and Ear Hospital, Melbourne, VIC 3002, Australia; 4Ophthalmology, Department of Surgery Melbourne, University of Melbourne, Melbourne, VIC 3000, Australia; 5Centre for Innate Immunity and Infectious Diseases, Hudson Institute of Medical Research, Melbourne, VIC 3168, Australia; 6Department of Molecular and Translational Science, Monash University, Melbourne, VIC 3168, Australia

**Keywords:** mitochondria, mitochondrial biogenesis, deoxyribonucleosides, ECHS1 deficiency, mitochondrial disease

## Abstract

The lack of effective treatments for mitochondrial disease has seen the development of new approaches, including those that stimulate mitochondrial biogenesis to boost ATP production. Here, we examined the effects of deoxyribonucleosides (dNs) on mitochondrial biogenesis and function in Short chain enoyl-CoA hydratase 1 (ECHS1) ‘knockout’ (KO) cells, which exhibit combined defects in both oxidative phosphorylation (OXPHOS) and mitochondrial fatty acid β-oxidation (FAO). DNs treatment increased mitochondrial DNA (mtDNA) copy number and the expression of mtDNA-encoded transcripts in both CONTROL (CON) and ECHS1 KO cells. DNs treatment also altered global nuclear gene expression, with key gene sets including ‘respiratory electron transport’ and ‘formation of ATP by chemiosmotic coupling’ increased in both CON and ECHS1 KO cells. Genes involved in OXPHOS complex I biogenesis were also upregulated in both CON and ECHS1 KO cells following dNs treatment, with a corresponding increase in the steady-state levels of holocomplex I in ECHS1 KO cells. Steady-state levels of OXPHOS complex V, and the CIII_2_/CIV and CI/CIII_2_/CIV supercomplexes, were also increased by dNs treatment in ECHS1 KO cells. Importantly, treatment with dNs increased both basal and maximal mitochondrial oxygen consumption in ECHS1 KO cells when metabolizing either glucose or the fatty acid palmitoyl-L-carnitine. These findings highlight the ability of dNs to improve overall mitochondrial respiratory function, via the stimulation mitochondrial biogenesis, in the face of combined defects in OXPHOS and FAO due to ECHS1 deficiency.

## 1. Introduction

Mitochondrial disease affects approximately 1 in 4800 live births, causing debilitating, multi-systemic metabolic disorders and significant morbidity and mortality [1]. Unfortunately, effective therapies for mitochondrial diseases are lacking, with symptomatic treatments being the only option for many patients. Most rely on mitochondrial ‘cocktails’ that include co-enzyme Q_10_ (CoQ_10_), creatine, L-carnitine, vitamins and other compounds that have only anecdotal efficacy [2]. While some individual case studies suggest that mitochondrial ‘cocktails’ can improve mitochondrial disease patient health, a Cochrane review of mitochondrial therapies found little supporting evidence for their use [3].

The lack of effective treatments has driven the development of new therapies to treat mitochondrial disease, with many clinical trials now underway [4]. These trials include KH176 (Sonlicromanol) [5] and EPI-743 [6], which have antioxidant properties, idebenone [7], riboflavin [8], pyruvate [9] and 5-aminovulinic acid [10], which are metabolites designed to boost OXPHOS function, and MTP-131 (elamipretide; SS-31), a molecule which protects mitochondrial inner membrane phospholipids [11].

Another potential avenue for mitochondrial disease treatment involves the stimulation of mitochondrial biogenesis, which can increase mitochondrial function in a non-specific manner. Stimulating mitochondrial biogenesis will increase mitochondrial mass, boosting ATP generation back above the critical disease threshold to alleviate disease symptoms [12]. Mitochondrial biogenesis is controlled by the peroxisomal proliferator activated receptor (PPAR) family of genes (alpha, beta and gamma), as well as PPARγ coactivator 1-alpha (PGC-1α), a transcription factor recognised as the ‘master regulator’ of mitochondrial biogenesis [13]. PGC-1α can induce the expression of nuclear respiratory factor 1 (NRF1) or NRF2, which then binds to the promoter of mitochondrial transcription factor A (TFAM) to increase expression [13]. TFAM then drives mitochondrial DNA replication to increase mtDNA copy number, as well as increasing mtDNA transcription and mitochondrial steady-state protein levels [14]. Consequently, an increase in mtDNA copy number is a useful marker of mitochondrial biogenesis stimulation [15].

Clinical trials to stimulate mitochondrial biogenesis have tested a range of compounds, including resveratrol, bezafibrate and sodium phenylbutyrate. Resveratrol targets the PGC-1α pathway through activation of SIRT1, and has been found to decrease oxidative stress in the lungs of chronic obstructive pulmonary disease (COPD) patients [16]. Bezafibrate is a pan-PPAR agonist approved for use in diabetic patients, and has recently been trialled in patients with the m.3243A > G *MTTL1* mutation [17]. Bezafibrate was found to reduce complex IV-deficient muscle fibres and improve cardiac function, but may not be suitable for long term treatment due to dysregulation of fatty acid metabolism [17]. Sodium phenylbutyrate has been used to treat patients with defects in ornithine transcarbamylase (OTC), a mitochondrial enzyme involved in the synthesis of citrulline, with measurable improvements in clinical symptoms associated with hyperammonemia and hyperglutaminemia [18].

The addition of deoxyribonucleosides (dNs) or deoxyribonucleotide triphosphates (dNTPs) has been shown to increase mtDNA (and stimulate mitochondrial biogenesis) in a range of different cell types [19]. DNs have been administered to patient derived cell lines deficient in deoxyguanosine kinase (dGK), and were found to incorporate into the cytosolic and mitochondrial salvage pathways in vitro [20] and restore mtDNA copy number [21,22]. DNs also increase mtDNA copy number and mtDNA repair in p53R2-deficient fibroblasts [23].

Additionally, dNs have been use therapeutically to stimulate an increase in mtDNA copy number in patients with mitochondrial thymidine kinase 2 deficiency, bypassing the defect in nucleotide generation that causes mtDNA depletion and pathogenesis in these patients [24]. As such, dNs may be a promising avenue for treating mitochondrial disease via the stimulation of mitochondrial biogenesis, which in turn will boost mitochondrial ATP output.

To test this, we have used dNs to stimulate mitochondrial biogenesis in 143BTK^−^ cells which lack Short chain enoyl-CoA hydratase 1 (ECHS1) expression following CRISPR/Cas9 editing of *ECHS1* (ECHS1 ‘knock out’) [25]. ECHS1 is a mitochondrial fatty acid β-oxidation (FAO) protein that catalyses the hydration of trans-Δ^2^-enoyl-CoA thioesters to 3-_L_-hydroxyacyl-CoA thioesters [26]. Patients deficient in ECHS1 (ECHS1 deficiency, ECHS1D) present as a specific subset of Leigh Syndrome [27], with developmental delay, dystonia and numerous neurological defects [28]. Interestingly, in addition to the ECHS1 deficiency which affects both FAO and amino acid metabolism [29], loss of ECHS1 expression has also been reported to disrupt OXPHOS activity and steady-state levels [30,31]. As such, ECHS1D is characterised by metabolic defects in both OXPHOS and FAO [25].

In this study, we examined the effects of dNs treatment in our previously described 143BTK^−^ ECHS1 ‘knock out’ (ECHS1 KO) cells, which exhibit a similar combined defect in both OXPHOS and FAO as observed in patient cell lines [25]. We found that dNs stimulate mitochondrial biogenesis, with increased mtDNA copy number in both CON and ECHS1 KO cells. This was associated with elevated expression of mtDNA-encoded transcripts, as well as altered global nuclear gene expression. Gene sets involved in mitochondrial OXPHOS were upregulated, including genes involved in OXPHOS complex I biogenesis. Correspondingly, steady-state levels of holocomplex I were increased in ECHS1 KO cells following dNs treatment, with increases in other OXPHOS complexes and the respiratory supercomplex also observed. Notably, this stimulation of mitochondrial biogenesis was associated with improved mitochondrial respiratory capacity in ECHS1 KO cells.

These findings highlight the ability of dNs to improve overall mitochondrial function in ECHS1 KO cells via the stimulation mitochondrial biogenesis, as well as their future potential for development into novel mitochondrial disease therapies where combined defects in OXPHOS and FAO are present.

## 2. Results

### 2.1. dNs Treatment Increases mtDNA Copy Number in Both Control and ECHS1 KO Cells

To stimulate mitochondrial biogenesis, un-edited 143BTK^−^ control cells (CON), or CRISPR/Cas9 *ECHS1* edited 143BTK^−^ cells (ECHS1 KO), were treated with a combination of four deoxyribonucleosides (dNs), containing 50 μM dG, 10 μM dC, 10 μM dA and 10 μM dT. These dNs concentrations are based on previously published studies and have been found to cause improvement without toxic side effects [32,33].

Following eight days of treatment, mtDNA copy number was assessed as a marker of mitochondrial biogenesis in both CON and ECHS1 KO (Figure 1A). Treatment with dNs increased mtDNA copy number in both CON (2-fold increase; *p* < 0.05) and ECHS1 KO (10-fold increase; *p* < 0.05) cells. MtDNA copy number was not significantly different between CON and ECHS1 KO cells before (*p* = 0.143), or after (*p* = 0.093) dNs treatment.

### 2.2. dNs Treatment Increases mtDNA Transcript Levels and Alters Global Gene Expression

Treatment of CON and ECHS1 KO cells with dNs for eight days resulted in significant changes to global transcript levels. Pearson correlation clustering of samples indicates a larger differential effect due to dNs treatment, compared to the difference between untreated CON and ECHS1 KO cells alone (Figure 1B). Transcriptomic profiling revealed 16,435 (8571 upregulated and 7864 downregulated) differentially regulated genes in CON cells after eight days of dNs treatment (FDR > 0.05), compared to 13,739 (6723 upregulated and 7016 downregulated) differentially regulated genes in ECHS1 KO cells after eight days of dNs treatment (Appendix A).

The mtDNA-encoded transcripts *MT-CO1* (3.24 log2 fold change in CON, 2.90 log2 fold change in ECHS1 KO), *MT-CO2* (3.35 log2 fold change in CON, 3.00 log2 fold change in ECHS1 KO), *MT-CO3* (3.25 log2 fold change in CON, 2.73 log2 fold change in ECHS1 KO), *MT-ND1* (3.43 log2 fold change in CON, 2.78 log2 fold change in ECHS1 KO), *MT-ND2* (3.62 log2 fold change in CON, 3.26 log2 fold change in ECHS1 KO), *MT-ND4* (3.10 log2 fold change in CON, 2.73 log2 fold change in ECHS1 KO) and *MT-CYB* (4.29 log2 fold change in CON, 3.70 log2 fold change in ECHS1 KO) were among the top genes with increased expression in both CON and ECHS1 KO cells following dNs treatment (Figure 1C). Conversely, no genes with known mitochondrial function were downregulated in the top 50 genes. Interestingly, genes such as *TFAM* and *POLG*, typically involved in the replication of mtDNA, were downregulated in both CON (*TFAM* −1.07 log2 fold change, *POLG* −0.28 log2 fold change) and ECHS1 KO (*TFAM* −0.46 log2 fold change, *POLG* −0.53 log2 fold change) after dNs treatment, suggesting other pathways are involved in stimulating mitochondrial biogenesis. A full list of differentially regulated genes and their log2 fold change values can be found in Appendix A.

The significant increases in mtDNA-encoded gene expression observed in the transcriptomic analysis were subsequently confirmed by qPCR (Figure 1D). CON and ECHS1 KO cells both exhibited significant increases in almost all mtDNA-encoded transcript levels examined, including *ND1* and *ND6* (complex I), *CYB* (complex III), *COX1* and *COX2* (complex IV) and *ATP6* and *ATP8* (complex V) (*p* < 0.05) when treated with dNs (Figure 1D). Mitochondrial rRNA genes *RNR1* and *RNR2* were both increased in CON cells (*p* < 0.01), but not in ECHS1 KO cells, following dNs treatment. Notably, dNs treatment in ECHS1 KO cells increased expression to levels similar or higher than of those observed in untreated CON cells (Figure 1D).

The expression of the nuclear-encoded OXPHOS complex I subunit genes *NDUFB11* (whose gene product is suggested to putatively interact with ECHS1 [25]) and *NDUFA1* (which is upregulated in both CON and ECHS1 KO cells) were also examined by qPCR. The expression of *NDUFB11* was decreased in ECHS1 KO cells (*p* < 0.05) but not in CON cells following dNs treatment, whereas expression of *NDUFA1* was increased in both CON and ECHS1 KO cells following dNs treatment (Figure 1E).

### 2.3. dNs Treatment Alters the Gene Expression of Multiple Cell Processes, Including Upregulation of Mitochondrial Metabolism

Transcriptomic analysis after eight days of dNs treatment detected 1483 reactome gene sets in CON cells, with 34 significantly upregulated and 895 significantly downregulated compared to untreated CON cells (Figure 2A). Similarly, 1479 reactome gene sets were identified in ECHS1 KO cells, with 147 significantly upregulated and 630 significantly downregulated by dNs treatment (Figure 2B). Included in the top 50 differentially regulated gene sets (ranked by effect size) following dNs treatment were many associated with mitochondrial function. This includes ‘tRNA processing in the mitochondria’ (*MT-CO1*, *MT-CO2*, *MT-CO3*, *MT-CYB*, *MT-ND1*, *MT-ATP6*, *MT-ATP8*) and ‘complex I biogenesis’ (*ACAD9*, *COA1*, *ECSIT*, *NDUFA1*, *MT-ND1-6*), which were the most upregulated in both CON and ECHS1 KO cells following treatment (Figure 2C,D).

Each of the six modules that assemble to form mature holocomplex I had increased expression of their constituent subunits, or their associated assembly factors, in both CON cells and ECHS1 KO cells following dNs treatment (Figure 2E). These include: *MT-ND1*, *NDUFA3* and *NDUFA8* in the ND1-module; *MT-ND2*, *MT-ND3*, *MT-ND4L* and *ACAD9* in the ND2-module; *MT-ND4*, *NDUFB1*, *NDUFB4* and *FOXRED1* in the ND4-module; *MT-ND5*, *NDUFB8* and *NDUFB3* in the ND5-module; *NDUFA2, NDUFV2*, *NDUFA7* and *NDUFAF2* in the N-module; and, *NDUFS8*, *NDUFA9*, *NDUFAF3* and *NDUFAF8* in the Q-module (Figure 2E).

Interestingly, the Q-module assembly factor *NUBPL* and the ND4-module assembly factors *TMEM70* and *ATP5SL* were downregulated following dNs treatment in both CON and ECHS1 KO cells. Conversely, the assembly factor *NDUFAF4*, which was previously found to be downregulated in ECHS1 KO cells [25], had increased expression following dNs treatment (Figure 2E).

Apart from these transcriptional changes associated with complex I and its assembly, ‘Transcriptional activation of mitochondrial biogenesis’ (*ACSS2*, *ALAS1*, *ATF2*), ‘Respiratory electron transport’ (*ACAD9*, *COA1*, *COQ10A*, *COQ10B*), and ‘ATP synthesis by chemiosmotic coupling’ (*ATP5F1A*, *ATP5F1B*) were also significantly upregulated in both CON and ECHS1 KO cells following dNs treatment (Appendix A). ECHS1 KO cells also had increased expression of the ‘Mitochondrial Fatty Acid Beta-Oxidation’ (*ACADVL*, *ACADL*, *ACAD8*, *ACAD11*) gene set. A full list of significant (as determined by FDR) differentially regulated gene sets, and their log2 fold change values can be found in Appendix A.

These findings suggest that dNs treatment is associated with an increase in the expression of multiple mitochondrial pathways, including mtDNA-encoded protein transcription and translation, complex I biogenesis and electron transport, which together may increase mitochondrial respiratory capacity.

### 2.4. dNs Stimulate Mitochondrial Biogenesis and Restore Function in ECHS1 KO Cells

To determine the effect of dNs treatment on whole gene sets, and to examine if dNs can improve specific defects found in ECHS1 KO cells, a multi contrast pathway enrichment analysis was performed as previously described [34]. This transcriptomic analysis compares the magnitude of change between untreated ECHS1 KO and CON cells, and compares that to the magnitude of change between untreated and dNs treated ECHS1 KO cells. From this comparison, it can be determined if reduced expression of a gene set in ECHS1 KO cells can be increased following dNs treatment. This analysis revealed that untreated ECHS1 KO cells have increased ‘mucopolysaccharidosis’ (0.72 s.dist), ‘Biotin transport and metabolism’ (0.56 s.dist), and ‘reduction of cytosolic calcium levels’ (0.43 s.dist) compared to untreated CON cells (Figure 3A left column, where red indicates higher expression in untreated ECHS1 KO cells compared to untreated CON cells). However, treatment with dNs decreased these effect sizes in ECHS1 KO cells (−0.26 s.dist, −0.29 s.dist and −0.50 s.dist respectively) (Figure 3A, right column, which compares the magnitude of change between untreated and dNs treated ECHS1 KO cells, with the magnitude of change between untreated CON and ECHS1 KO cells), restoring them to levels that are not significantly different from untreated CON cells (Appendix A).

Conversely, untreated ECHS1 KO cells have downregulated ‘complex I biogenesis’ (−0.12 s.dist), ‘selenoamino acid metabolism’ (−0.34 s.dist), and ‘mitochondrial iron-sulfur cluster biogenesis’ (−0.44 s.dist) compared to untreated CON cells (Figure 3A left column, shaded blue), yet these gene sets were upregulated in ECHS1 KO cells following dNs treatment (0.73 s.dist, 0.52 s.dist and 0.46 s.dist respectively) (Figure 3A right column, shown in red). ECHS1 KO cells also had downregulated ‘translation initiation complex formation’ (−0.51 s.dist) and ‘formation of a pool of free 40S subunits’ (−0.44 s.dist) compared to untreated CON cells that were upregulated in ECHS1 KO cells following dNs treatment (0.37 s.dist and 0.65 s.dist respectively).

A contour density plot of ‘complex I biogenesis’ gene set members (*ACAD9*, *COA1*, *ECSIT*, *NDUFA1*, *MT-ND1*, *MT-ND2*, *MT-ND3*, *MT-ND4*, *MT-ND5*, *MT-ND6*) highlights the discordant regulation due to dNs treatment (Figure 3B). Comparison between untreated ECHS1 KO cells and untreated CON cells is shown on the X axis, while comparison between untreated and dNs treated ECHS1 KO cells is shown on the Y axis. Gene members are primarily detected in the top left quadrant, representing their decreased expression in untreated ECHS1 KO cells compared to untreated CON cells (−0.12 s.dist), but increased expression in ECHS1 KO cells following dNs treatment (compared to untreated ECHS1 KO cells, 0.73 s.dist) (Figure 3B).

Similarly, a contour density plot showing discordant regulation of gene set members of ‘Respiratory electron transport’ (*COQ10A*, *COQ10B*, *COX5A*, *COX5B*, *ACAD9*) shows clustering of genes in the top left quadrant, again indicating downregulated expression in untreated ECHS1 KO cells compared to untreated CON cells (−0.16 s.dist), but with increased expression in ECHS1 KO cells following dNs treatment (0.59 s.dist) (Figure 3C). ‘Complex I biogenesis’ and ‘Respiratory electron transport’ were both upregulated following dNs treatment to levels above untreated CON (Appendix A).

A full list of comparatively significantly (as determined by FDR) differentially regulated gene sets sorted by standard deviation (SD) for discordant analysis [34] is available in Appendix A.

### 2.5. dNs Treatment Increases the Steady-State Levels of Individual OPXHOS Complex Subunits and Mature OXPHOS Holocomplexes in ECHS1 KO Cells

Of the OXPHOS complex subunits examined, only MTCO2 protein levels were increased in CON cells to 273 ± 94.3% of untreated CON levels (*p* < 0.05), with ATP5A (*p* = 0.131), UQCRC2 (*p* = 0.341), NDUFB8 (*p* = 0.384), VDAC1 (*p* = 0.180), and SDHA (*p* = 0.785) levels remaining unchanged following dNs treatment (Figure 4A, full Western blot images are shown in Appendix A).

After treatment of ECHS1 KO cells with dNs, the steady-state levels of NDUFB8 (~350%, *p* < 0.05) and MTCO2 (~450%, *p* < 0.05) were both increased compared to untreated ECHS1 KO cells, whereas ATP5A (*p* = 0.430), UCQRC2 (*p* = 0.738), VDAC1 (*p* = 0.211) and SDHA (*p* = 0.630) were not significantly different (Figure 4A). Notably, this increased the steady-state levels of NDUFB8 and MTCO2 above those observed in untreated CON cells (*p* < 0.05), indicating the ability of dNs treatment to restore the reduced levels of these OXPHOS subunits (Figure 4A).

The steady-state levels of the mature OXPHOS complexes were also examined by native gel electrophoresis (Figure 4B, full Western blot images are shown in Appendix A). After treatment with dNs, the steady-state levels of OXPHOS complexes I (*p* = 0.933), IV (*p* = 0.273), III_2_ (*p* = 0.097), V (*p* = 0.935), and the CIII_2_/CIV (*p* = 0.314) and CI/CIII_2_/CIV (*p* = 0.543) supercomplexes were unchanged in CON cells. However, in ECHS1 KO cells, steady-state levels of complex I (~300%, *p* < 0.05), complex III_2_ (~480%, *p* < 0.05), and complex V (~850%, *p* < 0.05) were increased compared to untreated ECHS1 KO cells following dNs treatment (Figure 4B). Furthermore, dNs treatment increased complex I and complex V steady-state levels to similar or above those in untreated CON cells. Steady-state levels of the CIII_2_/CIV (~550%, *p* < 0.05) and CI/CIII_2_/CIV (~1300%, *p* < 0.05) OXPHOS supercomplexes were also increased in ECHS1 KO cells compared to untreated ECHS1 KO following dNs treatment (Figure 4B). DNs treated samples also showed an increase in the steady-state levels of the CI/CIII_2_ supercomplex, as detected by the UQCRC1 (CORE1) antibody (Appendix A, panel ii).

Importantly, the reduced steady-state levels of the OXHPOS complexes and supercomplexes observed in ECHS1 KO cells were restored to similar levels (or higher) than those in untreated CON cells by dNs treatment.

### 2.6. Functional Analysis of dNs Treated Cells Reveals an Increase in Mitochondrial Respiratory Capacity

Mitochondrial respiration in the presence of the fatty acid palmitoyl-l-carnitine was increased after dNs treatment in both CON cells and ECHS1 KO cells (Figure 4C). The basal respiration rate in CON cells was increased to 584 ± 125% of untreated CON cell rates (*p* < 0.01) and the maximal respiration rate was increased to 634 ± 28% of untreated CON cell rates (*p* < 0.001) following dNs treatment (Figure 4C). DNs had a similar effect in ECHS1 KO cells, with the basal respiration rate increased to 626 ± 344% of untreated ECHS1 KO cell rates (*p* < 0.05) and the maximal respiration rate increased to 699 ± 359% of untreated ECHS1 KO cell rates (*p* < 0.05) (Figure 4C). The non-phosphorylating (‘Leak’) respiration rate was unchanged in both CON and ECHS1 KO cells, while spare respiratory capacity (maximal rate—basal rate) increased in both CON cells (*p* < 0.001) and ECHS1 KO cells (*p* < 0.05) after dNs treatment (Table 1). The cell respiratory control ratio (maximal rate/leak rate) was increased in both CON cells (*p* < 0.0001) and ECHS1 KO cells (*p* < 0.05) after dNs treatment (Table 1).

Notably, the reduced basal (*p* < 0.05) and maximal (*p* < 0.01) respiration rates in ECHS1 KO cells were increased after treatment with dNs to levels significantly higher than those observed in untreated CON cells.

When metabolising glucose, the basal respiration rate in ECHS1 KO cells was increased to 161 ± 53% (*p* > 0.05) and the maximal respiration rate was increased to 210 ± 16% (*p* > 0.0001) compared to untreated ECHS1 KO cells following dNs treatment (Figure 4D). Treatment with dNs did not cause any significant increase in basal or maximal respiration rates in CON cells (*p* = 0.859 and *p* = 0.098 respectively) (Figure 4D). Notably, dNs treatment increased respiratory function in ECHS1 KO cells to similar levels in untreated CON cells, restoring the deficit present in ECHS1 KO cells [25]. The non-phosphorylating (‘Leak’) respiration rate was decreased in CON cells (*p* < 0.05), but not in ECHS1 KO cells following dNs treatment, while spare respiratory capacity increased in ECHS1 KO cells (*p* < 0.0001) but not in CON cells (Table 1). The cell respiratory control ratio was unchanged in both CON and ECHS1 KO cells after dNs treatment in the presence of glucose (Table 1).

### 2.7. OXPHOS Complex I Assembly Is Increased in ECHS1 KO Cells Following dNs Treatment

To confirm if the increase in ‘complex I biogenesis’ gene set expression has a functional impact on complex I assembly, we performed radiolabelled in vitro import and assembly assays of the complex I subunit NDUFA9 (Figure 5A).

The amount of radiolabelled NDUFA9 assembled into mature complex I was not significantly different between untreated (Lane 1) and dNs treated CON cells (Lane 4) at 15 min (*p* = 0.397) or at 60 min (Lanes 2 and 5 respectively) (*p* = 0.868) (Figure 5A).

As previously observed [25], the amount of assembled NDUFA9 in untreated ECHS1 KO cells (Lane 8) was significantly lower than in untreated CON cells after 60 min (Lane 2) (*p* < 0.01). However, dNs treatment of ECHS1 KO cells increased the amount of assembled NDUFA9 after 15 min (Lane 10, *p* < 0.05) and 60 min (Lane 11, *p* < 0.05) of import (Figure 5A). This increased the amount of NDUFA9 assembled in dNs treated ECHS1 KO cells to similar levels observed in untreated CON cells (Figure 5C).

The mature, proteinase K (PK) resistant form of NDUFA9 (m) was detectable at similar levels in both CON and ECHS1 KO mitochondria following dNs treatment after 60 min of import (Figure 5B). This indicates that the increased amount of NDUFA9 assembled into complex I in ECHS1 KO cells treated with dNs is due to increased biogenesis, and not to increased processing and/or import of NDUFA9.

## 3. Discussion

Mitochondrial disease affects approximately 1 in 4800 live births and commonly impacts cardiac, skeletomuscular and central nervous systems [35]. These disorders are commonly caused by defects in either oxidative phosphorylation (OXPHOS) or mitochondrial fatty acid β-oxidation (FAO) [36]. However, some mitochondrial disease aetiologies exhibit a combination of defects in both OXPHOS and FAO, including Long-chain 3-hydroxyacyl-CoA dehydrogenase (LCHAD), Medium-chain acyl-CoA dehydrogenase (MCAD) [37] and acyl-CoA dehydrogenase family member 9 (ACAD9) deficiencies [38,39,40]. Treatment options for these mitochondrial diseases are limited, with many having only anecdotal evidence of efficacy [3].

Stimulation of mitochondrial biogenesis is being explored as a possible treatment option for mitochondrial disorders, with the intention of increasing the total mitochondrial mass to restore ATP generation back above disease thresholds [41]. This may be particularly effective for diseases associated with heteroplasmic mtDNA mutations [42], or those which exhibit combined defects in OXPHOS and FAO, where residual mitochondrial function persists (albeit reduced below the normal range). Many compounds have been trialled to stimulate mitochondrial biogenesis, and although promising, have exhibited various limitations, including poor half-life [43] and lack of clinical efficacy [44]. We have recently described the stimulation of mitochondrial biogenesis with a combination of deoxyribonucleosides (dNs) and pioglitazone, and found that it increases mitochondrial respiratory capacity in MELAS 3243A > G cybrid cells [42]. DNs have also been shown to stimulate mitochondrial biogenesis independently of pioglitazone [20], and therefore may be suitable as a single agent for boosting mitochondrial function in disease settings.

While the exact mechanism involved is not clear, dNs treatment has been shown to increase mtDNA copy number in cells deficient in deoxyguanosine kinase, thymidine induced mtDNA depletion, or POLG deficiency [20,32]. dNs have been suggested to increase the available dNTP pool and the activity of the mitochondrial dNTP salvage pathway, with corresponding increases in thymidine kinase 2 (TK2) and dGK activity [45]. This increase in available dNTPs stimulates mtDNA replication, which in turn triggers mitochondrial biogenesis.

In this study, we examined the effects of dNs using an in vitro model of ECHS1 deficiency that exhibits both FAO and OXPHOS defects [25] and found that dNs treatment increased mtDNA copy number in both CON and ECHS1 KO cells. Furthermore, transcriptomic profiling identified increased expression of genes involved in mitochondrial biogenesis, in particular OXPHOS complex I biogenesis, which are key for increasing mitochondrial function.

DNs treatment increased the expression of mitochondrially encoded genes, in agreement with mitochondrial biogenesis stimulation. Specifically, expression of *MT-ND1*, *MT-ND2*, *MT-ND3*, *MT-ND4*, *NDUFA1* and *NDUFA11* (from the ‘complex I biogenesis’ gene set) was increased following dNs treatment in ECHS1 KO cells, with the expression of *MT-ND1* and *MT-ND6* increased to levels the same as in untreated CON cells.

Both CON and ECHS1 KO cells had similar upregulation of reactome pathways associated with mitochondrial function as a result of dNs treatment. This includes ‘tRNA processing in the mitochondrion’, ‘complex I biogenesis’ and ‘rRNA processing in the mitochondrion’. Furthermore, ‘Respiratory electron transport, ATP synthesis by chemiosmotic coupling, and heat production by uncoupling proteins’ were also found to be upregulated in ECHS1 KO cells following dNs treatment, suggesting a global upregulation of mitochondrial energy generation pathways in these cells.

Importantly, some of these pathways, which are downregulated in untreated ECHS1 KO cells, were increased to levels greater than in untreated CON cells by dNs treatment (see Figure 3A). This includes ‘complex I biogenesis’ and ‘mitochondrial iron-sulfur cluster biogenesis’. dNs treatment of ECHS1 KO cells and CON cells also increased the similarity of their gene expression patterns, as compared to untreated ECHS1 KO and CON cells. In the Pearson correlation heatmap, the untreated samples are visibly distinct, whereas the dNs treated samples are less divergent as the gene expression differences become smaller.

The increase in the expression of genes associated with ‘complex I biogenesis’ was evidenced by an increase in the steady-state levels of complex I and the OXPHOS CI/CIII_2_/CIV supercomplex in ECHS1 KO cells following dNs treatment. Steady-state levels of mature complex V were also increased, suggesting dNs treatment stimulates the biogenesis of not just complex I, but also other complexes of the respiratory chain.

The increase in ECHS1 KO steady-state complex I levels were associated with a functional increase in complex I biogenesis, as would be expected from the transcriptomics data. The assembly of the complex I subunit NDUFA9 into mature complex I was increased in ECHS1 KO cells by dNs treatment. In fact, the amount of assembled NDUFA9 was increased to levels similar to those found in untreated control cells. These results suggest that dNs treatment can indeed induce complex I biogenesis in the face of a combined OXHPOS-FAO defect as found in ECHS1 KO cells.

Interestingly, genes such as *CLPX* and members of the *LON* family (common mitochondrial proteases) are downregulated following dNs treatment in both CON and ECHS1 KO cells. This may also contribute to the increase in complex I (and other OXPHOS complex) steady-state levels, as mitochondrial protein turnover will be dampened. However, the mechanism that links dNs supplementation with mitochondrial proteostasis is unknown.

Overall, dNs treatment stimulated mitochondrial biogenesis (with a notable increase in complex I assembly), resulting in increased respiratory capacity, with both basal and maximal respiration rates in ECHS1 KO cells restored to untreated control levels. Furthermore, in the presence of palmitoyl-L-carnitine, spare respiratory capacity and the cell respiratory control ratio were both increased by dNs treatment, indicating an improved ability to respond to ATP demand.

Whilst dNs treatment increased mitochondrial biogenesis, commonly involved transcription factors, such as NRF1, NRF2 and PGC1-α, were not upregulated, suggesting non-classical pathways are instead induced [in fact, NRF1 and NRF2 (NFE2L2) were downregulated in both CON and ECHS1 KO cells following dNs treatment (Appendix A)]. Alternatively, it is possible that an increase in the free dNs pool within the mitochondria directly increases mtDNA transcription and subsequently mitochondrial biogenesis, without a corresponding increase in the expression of classical nuclear transcription factors. This results in increased mtDNA-encoded proteins levels, which triggers the assembly of de novo OXPHOS complexes to facilitate an increase in respiratory capacity. Indeed, others have shown that dNs can directly induce an increase in mtDNA replication and transcription which stimulates mitochondrial biogenesis [23], in particular when restoring balance in mtDNA depletion disorders [46,47]. Further studies may address the mechanism that drives this increase in mitochondrial biogenesis in more detail, as dNs do not appear to be stimulating mitochondrial biogenesis via classical pathways in our ECHS1 KO cells. This mechanism will need to be defined if future therapeutic investigations are to be undertaken. Furthermore, dNs may have different effects in vivo, and so pre-clinical testing in a suitable model of ECHS1 deficiency is required for future mechanistic studies and determination of efficacy.

In conclusion, we have found that dNs stimulate mitochondrial biogenesis in ECHS1-deficient cells, restoring mitochondrial respiratory capacity to within normal levels. These findings highlight the ability of dNs to increase mitochondrial function back above critical disease thresholds, as well as their potential for development into novel mitochondrial disease therapies where defects in both OXPHOS and FAO are present.

Abbreviations: ECHS1, Short chain enoyl-CoA hydratase 1; ECHS1D, ECHS1 Deficiency; FAO, mitochondrial fatty acid β-oxidation (FAO); LS, Leigh Syndrome; OXPHOS, oxidative phosphorylation.

## 4. Methods

### 4.1. Stock Solutions

Stock solutions were prepared as follows: 50 mM 2′-deoxyguanosine monohydrate (dG, Sigma-Aldrich, St Louis, MO, USA) in 50% *v*/*v* DMSO; 10 mM 2′-deoxycytidine (dC, Sigma-Aldrich) in distilled H_2_O; 10 mM 2′-deoxyadenosine monohydrate (dA, Sigma-Aldrich) in distilled H_2_O; and 10 mM 2′ deoxythymidine (dT, Sigma-Aldrich) in distilled H_2_O.

### 4.2. Cell lines and Culture Conditions

Cells were cultured in supplemented Dulbecco’s Modified Eagle Medium (DMEM) media containing 10% (*v*/*v*) fetal bovine serum (FBS), 50 units/mL penicillin and 50 μg/mL streptomycin at 37 °C/5% CO_2_. Cells were treated with a combination of four deoxyribonucleosides (dNs) containing 50 μM dG, 10 μM dC, 10 μM dA, and 10 μM dT at 37 °C/5 % CO_2_ for eight days in supplemented DMEM media with fresh media supplied every second day. Untreated (ut) controls were incubated with DMSO only at the relevant solute concentrations. Cell counts were performed by haemocytometer.

### 4.3. mtDNA Copy Number Analysis

mtDNA copy number was determined as described, by calculating the ratio of mtDNA to nuclear DNA (*ACTB*) per cell in relation to standard curves of known concentrations, ranging from 10^−1^ ng/μL to 10^−8^ ng/μL [42]. Quantitative PCR was performed on a Rotor Gene RG-3000 (Corbett Research, Mortlake, Australia), with RotorGene software (v6.0) used for data acquisition and analysis. Standard curves were used with a correlation coefficient of R^2^ > 0.9 and an efficiency coefficient of R > 0.95. Mitochondrial DNA copy number per cell was calculated as follows:


$$mtDNA\;copies\;\left(N_{mtDNA}\right)=\frac{\left(\mathrm{ng}/\mu L\right)\times \left(6.023\times 10^{14}\right)}{152\;\mathrm{bp}\times 660}$$



$$\beta -globin\;copies\;\left(N_{\beta -globin}\right)=\frac{\left(\mathrm{ng}/\mu L\right)\times \left(6.023\times 10^{14}\right)}{268\;\mathrm{bp}\;\times 660}$$



$$mtDNA\;copy\;number\;per\;cell=\frac{N_{mtDNA}}{2\times N_{\beta -globin}}$$


### 4.4. Measurement of Oxygen Consumption Rates

High-resolution respirometry was performed with an Oxygraph-2K oxygen electrode (Oroboros, Innsbruck, Austria). Basal respiration in intact cells was measured in DMEM GlutaMAX or in glucose free media supplemented with 1 mM Sodium Pyruvate, 40 μM Palmitoyl-L-Carnitine, 1 mM GlutaMAX, 10% dialysed FBS. Non-phosphorylating respiration (‘Leak’) was measured in the presence of 5 mg/mL oligomycin and maximal respiration determined by the sequential addition of 1 μM aliquots of carbonyl cyanide-4-(trifluoromethoxy)phenylhydrazone (FCCP). Non-mitochondrial respiration was measured in the presence of 2 μM antimycin A. Spare respiratory capacity (maximal rate-basal rate) and cell respiratory control ratios (maximal/proton leak) were calculated according to [48], using Data software (version 4.51, Oroboros Instruments, Innsbruck, Austria) and expressed as pmol O_2_/s/mg of whole cell protein.

### 4.5. Denaturing Gel Electrophoresis

Proteins was separated using denaturing gel electrophoresis, as previously described [49]. In brief, 40 μg of total cell protein or of isolated mitochondria was separated on a 10–16% (*w*/*v*) Tris-tricine continuous gradient gel at 100 V/25 mA for approximately 14 h.

### 4.6. Native Gel Electrophoresis

Blue-native polyacrylamide gel electrophoresis (BN-PAGE) was performed as previously described [49]. In brief, 50 μg of total cell protein was solubilised for 30 min on ice in 50 μL of 20 mM Bis-Tris (pH 7.4), 50 mM NaCl, and 10% *(v*/*v*) glycerol containing either 1% (*v*/*v*) Triton X-100 (Sigma-Aldrich) or 1% (*w*/*v*) digitonin (Merck, Branchburg, NJ, USA). Samples were spun at 20,000× *g* for 5 min, 4 °C to pellet insoluble material, and the supernatant was combined with 5 μL of 10 × BN-PAGE loading dye (5% (*w*/*v*) Coomassie Blue G, 500 mM Ɛ-amino-n-caproic acid). Samples were resolved on a 4–13% (*w*/*v*) BN-PAGE gel at 100 V/7 mA for approximately 14 h at 4 °C.

### 4.7. Western Blotting

Semi-dry Western transfer and immune detection was performed as previously described [50]. Proteins were transferred to a PVDF membrane using a semi-dry transfer method, blocked in 10% (*w*/*v*) skim milk in 1xPBS/0.05% (*v*/*v*) Tween 20, then probed overnight with primary antibodies at 4 °C. Membranes were then incubated with appropriate HRP-coupled secondary antibody, and proteins visualised using ECL (GE Healthcare, Cincinnati, OH, USA). Proteins were visualized with a Chemidoc XRS imaging system (Bio-rad, Hercules, CA, USA). Primary antibodies used were against mitochondrial oxidative phosphorylation (OXPHOS) proteins NDUFB8, SDHB, UQCRC2, MTCO2 and ATP5A (OXPHOS ‘cocktail’, Abcam, Cambridge, UK, ab110411), and VDAC1 (Abcam, ab14734). For Native PAGE, the following antibodies were used: SDHA (CII, Abcam, ab14715), ATP5A (CV, Abcam, ab14748), UQCRC1 (CIII_2_, CIII_2_/CIV, Abcam, ab110252), MT-COI (CIV, Abcam, ab14705), and NDUFA9 (CI, CI/CIII_2_/CIV, raised in rabbits, as previously described [51]). Protein band intensities were calculated using ImageJ software (National Institutes of Health, Bowie, MD, USA) from three independent, non-saturated images. Protein complex levels were standardised to SDHA for SDS-PAGE analysis and to mature holocomplex II for BN-PAGE analysis. Significant differences were determined using Student’s two-tailed *t*-tests.

### 4.8. RNA Sequencing and Differential Expression Analysis

Total RNA was extracted using the RNeasy Plus Kit with on-column DNA elimination (Qiagen, Hilden, Germany) according to manufacturer’s instructions. Transcriptome-wide mRNA sequencing was performed by the Australian Genome Research Facility (Melbourne, Australia) from four independent experiments (n = 4). RNA purity and integrity were confirmed by BioAnalyser (Agilent, Santa Clara, CA, USA). Libraries were prepared using the Ribo-zero Gold protocol (Illumina, San Diego, CA, USA) and assessed by Bioanalyser DNA 1000 chip (Agilent). Libraries were pooled and clustered through the Illumina cBot system using TruSeq PE Cluster Kit v3, followed by sequencing on the Illumina HiSeq 2500 system with TruSeq SBS Kit v3 reagents. 100 bp single-end reads were produced with a depth of at least 30 million reads per sample. The primary sequence data was generated using the Illumina bcl2fastq 2.18.0.12 pipeline, with per base sequence quality for all samples >94% bases above Q30. Skewer [v0.2.2 [52]] was used to trim low quality bases from the 3′ end of reads. Trimmed reads were then mapped to the Gencode (v37) human transcriptome using Kallisto [53]. Data were read into R, followed by aggregation of transcript level counts to gene level count data for downstream analysis.

Clustering between the samples was inspected using multidimensional scaling analysis in R. DESeq2 [54] version 1.32.0 was used to perform multi comparison differential gene expression analysis between untreated and dNs treated control 143BTK^−^ cells and 143BTK^−^ ECHS1 ‘knockout’ cells. DESeq2 data underwent multi-contrast gene set analysis with mitch v1.4.0 with the default parameters [34]. Mitch applies a rank-ANOVA statistical approach to detect groups of genes with altered expression in one or more contrasts. Gene sets used in the analysis were downloaded from Reactome (accessed 10 December 2021) [55]. DESeq2 and mitch results with FDR < 0.05 were considered statistically significant.

Topographical mapping of RNA-Seq log2 fold-changes to complex I (PDB: 5LDW) and complex I assembly factors (AlphaFold 2.0: ACAD9 (AF-Q9H845), AIF (AF-O95831), ATP5SL (AF-Q9NW81), C9orf72 (AF-Q96LT7), COA1 (AF-Q9GZY4), DMAC1 (AF-Q96GE9), ECSIT (AF-Q9BQ95), FOXRED1 (AF-Q96CU9), NDUFAF1 (AF-Q9Y375), NDUFAF2 (AF-Q8N183), NDUFAF3 (AF-Q9BU61), NDUFAF4 (AF-Q9P032), NDUFAF5 (AF-Q5TEU4), NDUAF6 (AF-Q330K2), NDUFAF7 (AF-Q7L592), NDUFAF8 (AF-A1L188), NUBPL (AF-Q8TB37), SFXN4 (AF-Q6P4A7), TMEM70 (AF-Q9BUB7), TMEM126B (AF-Q8IUX1) [56,57]) was conducted with python scripts generously provided by Dr David Stroud (Bio21) and as previously described [58].

### 4.9. cDNA Synthesis

First strand synthesis was achieved using random primers at 70 °C for 5 min. dNTPs and M-MLRVT (H-) (M3682, Promega, Madison, WI, USA) were then added and incubated at 21 °C for 10 min, 42 °C for 50 min and then 70 °C for 15 min.

### 4.10. Real-Time Quantitative PCR (qPCR) Analysis

All qPCR reactions were carried out using cDNA purified as described above, with relative cDNA quantitation calculated using the ΔΔCt method from four independent samples. QPCR analysis of nuclear-encoded genes was performed using a Rotor-gene 3000 (Corbett Research) as described [45] with pre-designed KiCQSTART primer pairs (KSPQ12012G, Merck). B2M (identified as having a low coefficient of variation by differential expression analysis) was used as the calibrator/reference gene [59].

QPCR of mtDNA-encoded transcripts was performed using TaqMan Fast Advanced master mix (Thermo Fisher Scientific, Waltham, MA, USA) and TaqMan gene expression assays (Thermo Fisher Scientific) for MT-ND1 (Hs02596873), MT-ND6 (Hs02596879_g1), MT-CYB (Hs02596867), MT-CO1 (Hs02596864), MT-CO2 (Hs02596865_g1), ATP6 (Hs02596862_g1), ATP8 (Hs02596863_g1) RNR1 (Hs02596859_g1), RNR2 (Hs02596860_s1) and B2M (Hs00984230_m1) according to the manufacturer’s instructions.

### 4.11. In vitro Mitochondrial Import Assays

cDNA encoding NDUFA9 was cloned into the pGEM4Z vector (Promega, Madison, WI, USA) and protein translated using the TnT Coupled Reticulocyte Lysate System (Promega) in the presence of [35S]-methionine/cysteine. Translation products were incubated with freshly isolated mitochondria in 250 mM sucrose, 80 mM potassium acetate, 5 mM magnesium acetate, 10 mM sodium succinate, 1 mM dithiothreitol, 5 mM ATP and 20 mM HEPES pH 7.4 at 37 °C for the times indicated. Dissipation of the mitochondrial membrane potential (Δψ_m_) was performed in the presence of 10 μM FCCP (with no ATP or sodium succinate). Samples subjected to protease treatment were incubated on ice for 10 min with 100 μg/mL proteinase K (Sigma) before treatment with 1 mM PMSF for 10 min. 40 μg of each sample was resolved on either SDS–PAGE or BN–PAGE as described, with proteins transferred to PVDF membranes before exposure to storage phosphor screens (GE Healthcare) and detection using a Typhoon Laser Scanner (GE Healthcare). Protein band intensities were calculated using ImageJ (NIH) software from at least three independent experiments, normalised to the maximum amount of imported NDUFA9 after 60 min in controls, and converted to % of control levels at 60 min.

## Figures and Tables

**Figure 1 ijms-23-12610-f001:**
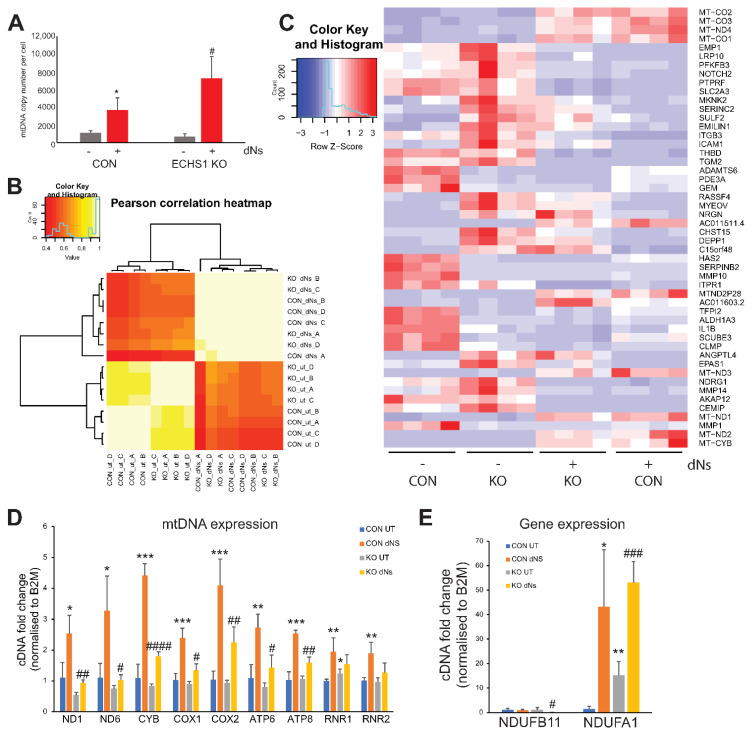
dNs treatment increased mtDNA copy number and alters global gene expression. (**A**) MtDNA copy number was increased in both control (CON) and ECHS1 KO cells after 8 days of dNs treatment. (**B**) Pearson correlation heatmap of transcriptome expression in CON cells and ECHS1 KO (KO) cells, either untreated (ut) or treated with dNs for 8 days. (**C**) Heatmap of 50 most significantly differentially expressed genes across all samples. The expression of mtDNA-encoded transcripts was increased in both CON and ECHS1 KO cells following dNs treatment. (**D**) Expression of mtDNA-encoded transcripts was increased following dNs treatment in both CON and ECHS1 KO cells. (**E**) Expression of nuclear-encoded OXPHOS complex I subunit genes were increased in both CON and ECHS1 KO cells following dNs treatment. Data shown is mean ± standard deviation (s.d.), with n ≥ 3. * *p* < 0.05, ** *p* < 0.01, *** *p* < 0.001 compared to untreated (ut) CON cells; # *p* < 0.05, ## *p* < 0.01, ### *p* < 0.001, #### *p* < 0.0001 compared to untreated (ut) ECHS1 KO cells (Student’s two-tailed *t* tests).

**Figure 2 ijms-23-12610-f002:**
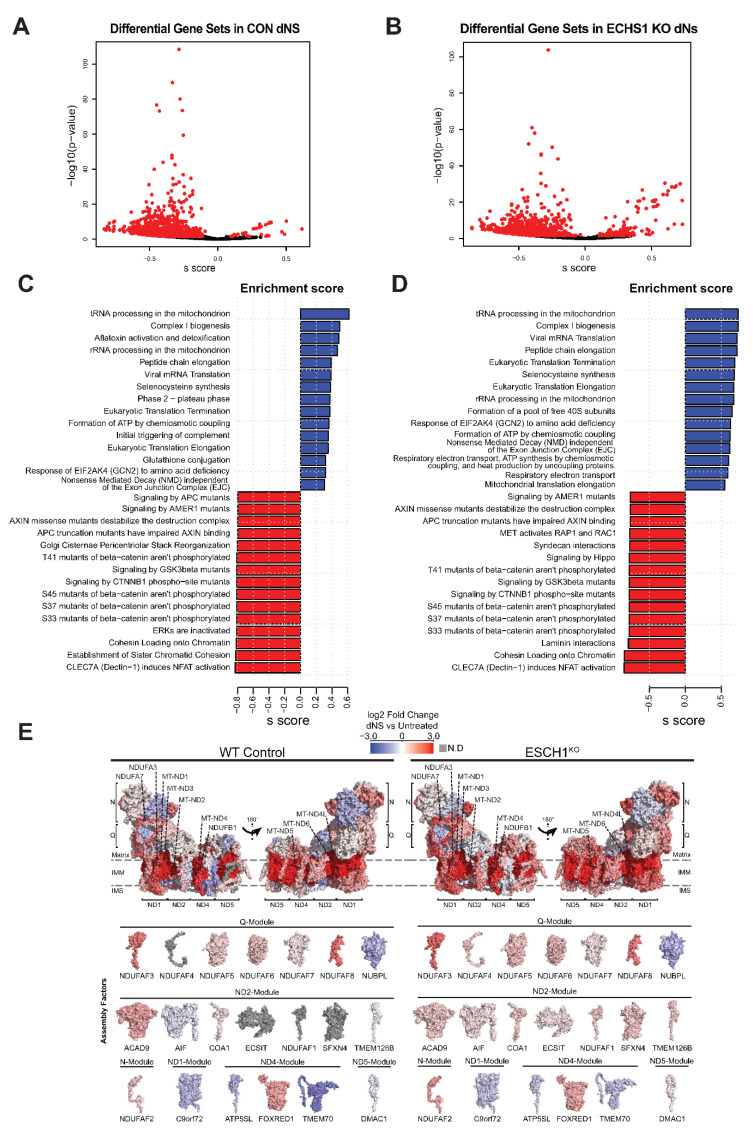
dNs treatment alters the gene expression of multiple cellular process and increases complex I biogenesis. Volcano plot depicting gene expression changes in (**A**) CON cells or (**B**) ECHS1 KO cells following dNs treatment. 1483 gene sets were identified in CON cells, with 34 upregulated and 895 downregulated after dNs treatment (shown in red, FDR < 0.05). Similarly, 1479 gene sets were identified in ECHS1 KO cells, with 147 upregulated and 630 downregulated (shown in red, FDR < 0.05). (**C**) Bar plot of the top 15 upregulated and downregulated gene sets in CON cells due to dNs treatment, as ranked by effect size (s.dist). (**D**) Bar plot of the top 15 upregulated and downregulated gene sets in ECHS1 KO cells due to dNs treatment, as ranked by effect size (s.dist). (**E**) Topographical heatmaps showing RNA-Seq log_2_ fold-changes mapped onto the structures of complex I subunits and assembly factors for CON (left) and ESCH1 KO (right) cells treated with deoxyribonucleosides (dNs) relative to untreated cells. Complex I assembly factors are grouped according to the complex I module they are primarily involved with. Grey subunits were not detected (N.D).

**Figure 3 ijms-23-12610-f003:**
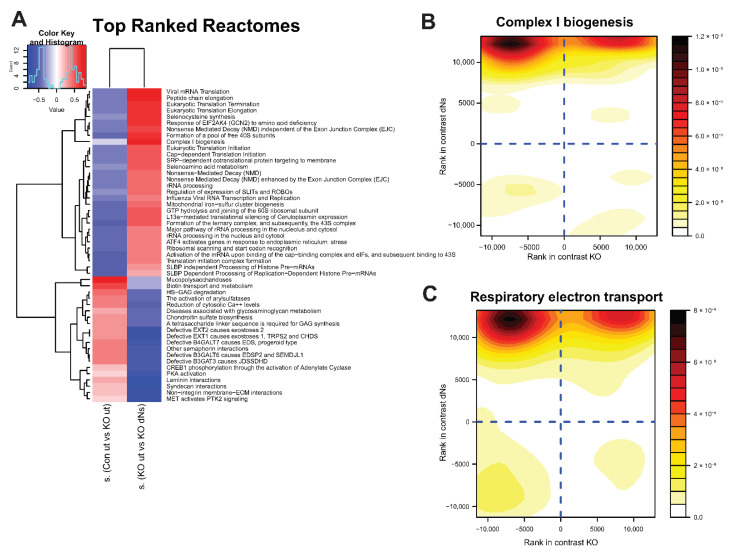
dNs stimulate genes involved in complex I biogenesis and respiratory electron transport in ECHS1 KO cells. (**A**) Heat map showing enrichment scores of gene sets selected by discordance between comparison of untreated control cells (CON ut) and untreated ECHS1 KO (ut) cells (left column), with comparison of untreated ECHS1 KO cells (ut) and dNs treated ECHS1 KO cells (dNs) (right column), as ranked by effect size (s.dist). Blue indicates gene sets that are downregulated, red indicates gene sets that are upregulated. Complex I biogenesis is a key gene set that was downregulated in ECHS1 KO cells (blue, left column) but increased following dNs treatment (red, right column). (**B**) Contour density plot showing discordant regulation of gene set members of complex I biogenesis. X axis shows rank in comparison between untreated ECHS1 KO cells and untreated CON cells, Y axis shows rank in comparison between untreated and dNs treated ECHS1 KO cells. Shading represents density of genes within section of graph. Darker colours indicate higher intensity. (**C**) Contour density plot showing discordant regulation of gene set members of respiratory electron transport. X axis shows rank in comparison between untreated ECHS1 KO cells and untreated CON cells, Y axis shows rank in comparison between untreated or dNs treated ECHS1 KO cells. Shading represents density of genes within section of graph. Darker colours indicate higher intensity.

**Figure 4 ijms-23-12610-f004:**
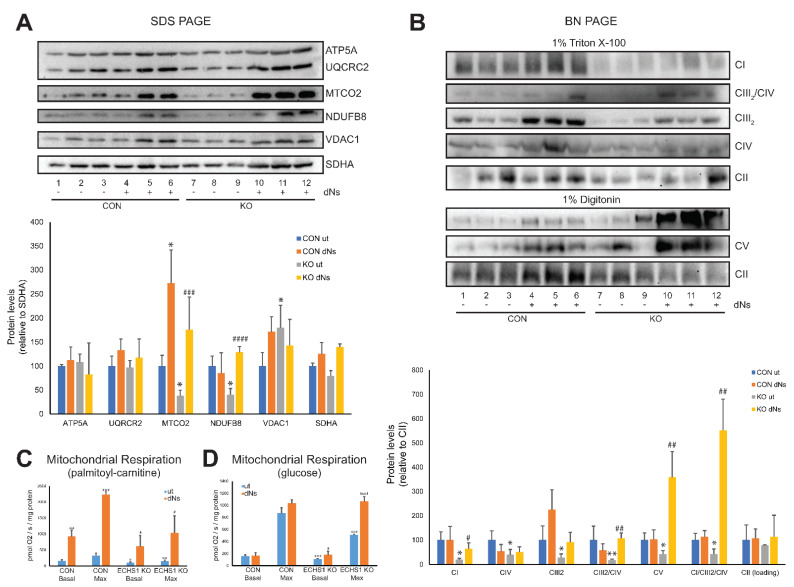
dNs treatment increases steady-state OXPHOS complex levels and mitochondrial respiratory capacity. (**A**) SDS-PAGE reveals that steady-state levels of NDUFB8 (complex I) and MTCO2 (complex IV) are decreased in untreated ECHS1 KO cells compared to untreated control CON cells, but increase following eight days of dNs treatment to control levels. (**B**) BN-PAGE reveals that steady-state levels of OXPHOS complexes I, III_2_, IV and the CIII_2_/CIV and CI/CIII_2_/CIV supercomplexes are reduced in untreated ECHS1 KO cells compared to untreated CON cells. Treatment with dNs increased the levels of complex I, complex V, and the CIII_2_/CIV and CI/CIII_2_/CIV supercomplexes in ECHS1 KO cells to levels similar or above those in untreated CON cells. (**C**) Mitochondrial respiration rates using palmitoyl-L-carnitine as substrate. Untreated ECHS1 KO cells had reduced basal and maximal respiration rates compared to untreated CON cells. dNs treatment increased both rates in ECHS1 KO cells to levels above those in untreated CON cells. (**D**) Mitochondrial respiration rates using glucose as substrate. Untreated ECHS1 KO cells had reduced basal and maximal respiration rates compared to CON cells. dNs treatment increased both rates in ECHS1 KO cells to levels similar to CON cells. Data shown is mean ± s.d., with n ≥ 3. * *p* < 0.05, ** *p* < 0.01, *** *p* > 0.001 compared to untreated (ut) CON cells; # *p* < 0.05, ## *p* > 0.01, ### *p* > 0.001, #### *p* > 0.0001 compared to untreated (ut) ECHS1 KO cells (Student’s two-tailed *t* tests). Images shown are representative of three independent experiments.

**Figure 5 ijms-23-12610-f005:**
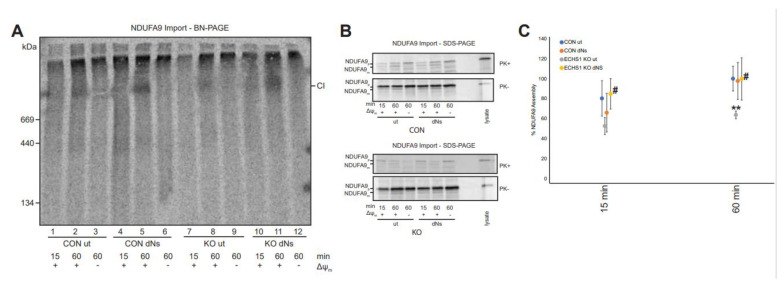
dNs treatment increases NDUFA9 assembly into complex I in ECHS1 KO cells. (**A**) BN-PAGE showing the assembly of NDUFA9 into mature complex I (CI) following solubilisation of 40 μg mitochondria per lane in 1% TX-100. (**B**) SDS-PAGE showing NDUFA9 following import into both untreated and dNs treated CON and ECHS1 KO mitochondria. NDUFA9 is detectable in its precursor (p) form and as a proteinase K (PK+) resistant mature (m) form. (**C**) Quantitation of NDUFA9 assembly into mature complex I after 15 and 60 min of import, normalised to maximum amount of NDUFA9 imported after 60 min in untreated CON. The amount of NDUFA9 assembled into complex I (CI) after 60 min was restored in dNs treated ECHS1 KO mitochondria to levels similar to untreated CON. ** *p* < 0.01 compared to untreated (ut) CON cells; # *p* < 0.05 compared to untreated (ut) ECHS1 KO cells (Student’s two-tailed *t* tests).

**Table 1 ijms-23-12610-t001:** Respiration rates in untreated and dNs treated control and ECHS1 KO cells.

Substrate		Basal pmol O_2_/s/mg	Maximal pmol O_2_/s/mg	Proton Leak pmol O_2_/s/mg	Spare Respiratory Capacity	Cell Respiratory Control Ratio
Palmitoyl-L-Carnitine	CON UT	157 ± 37	326 ± 74	65 ± 14	169 ± 38	5 ± 1
	CON + dNs	917 ± 197 **	2230 ± 93 ***	56 ± 10	1312 ± 290 ***	40 ± 2 ****
	ECHS1 KO UT	98 ± 30 *	148 ± 37 **	67 ± 10	51 ± 13 **	2 ± 1 **
	ECHS1 KO + dNs	612 ± 336 ^#^	1036 ± 532 ^#^	74 ± 6	424 ± 264 ^#^	14 ± 7 ^#^
Glucose	CON UT	160 ± 17	891 ± 86	132 ± 28	726 ± 74	7 ± 1
	CON + dNs	165 ± 49	1031 ± 97	53 ± 35 *	851 ± 75	31 ± 27
	ECHS1 KO UT	111 ± 7 ***	513 ± 16 ***	88 ± 54	404 ± 19 ****	9 ± 9
	ECHS1 KO + dNs	180 ± 59 ^#^	1065 ± 80 ^###^	63 ± 39	892 ± 59 ^####^	21 ± 11

CON, control cells; ECHS1 KO, ECHS1 KO cells; UT, untreated; dNs, deoxyribonucleosides. Spare respiratory capacity equals the maximal rate-basal rate; the Cell respiratory control ratio equals the maximal rate divided by the proton leak rate.* *p* < 0.05, ** *p* < 0.01, *** *p* > 0.001, **** *p* > 0.0001 compared to untreated (ut) CON cells; # *p* < 0.05, ### *p* > 0.001, #### *p* > 0.0001 compared to untreated (ut) ECHS1 KO cells (Student’s two-tailed *t* tests).

## Data Availability

The transcriptomic data discussed in this publication have been deposited in NCBI’s Gene Expression Omnibus and are accessible through GEO Series accession number GSE200252 study at: https://www.ncbi.nlm.nih.gov/geo/query/acc.cgi?acc=GSE200252, accessed on 10 April 2022.

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
