# Peer review of "Stimulating Mitochondrial Biogenesis with Deoxyribonucleosides Increases Functional Capacity in ECHS1-Deficient Cells"

_ijms, 2022, doi:10.3390/ijms232012610_

Round 1

Author Response

Dear Editors,

We thank the reviewers for their insightful comments, which we have now addressed in full. This includes additional text and changes to the figures. Below is a detailed point-by-point description of our response, which we feel has made our manuscript stronger overall.

Kind regards,

Matthew McKenzie PhD

Reviewer 1

  1. The model of this paper is too similar to that of previously published article “Pioglitazone and Deoxyribonucleoside Combination Treatment Increases Mitochondrial Respiratory Capacity in m.3243A>G MELAS Cybrid Cells”. What are the innovations and advantages?

Response: We thank the reviewer for their insight and thorough understanding of the literature. While there are some similarities with our previously published paper, this paper describes the ability of deoxyribonucleosides to restore mitochondrial function in cells that have lost expression of a nuclear-encoded gene (ECHS1) that results in a combined OXPHOS/FAO defect associated with Leigh syndrome. This is in contract to our previous work, where we used deoxyribonucleosides in combination with pioglitazone to increase mitochondrial function in cells with a mtDNA mutation associated with MELAS. As such, the compounds used, the genetic defect, and the associated disease are all different. This paper also uses high throughput data analysis, including RNAseq, to examine global expression changed due to deoxyribonucleoside treatment. This provides additional insight to the cellular changes that are induced by deoxyribonucleosides.

  1. The pixel of pictures are low and can't be seen clearly.

Response: We thank the reviewer for this feedback. Our initial submission included TIF images, which were low resolution. We now provide adjusted, high-resolution figures that are much clearer.

  1. In the conclusion section, it is recommended that the authors need to discuss the limitations of study.

Response:  We thank the reviewer for this suggestion. We have included the following text in the Discussion which outlines some of the limitations of the study:

‘Further studies may address the mechanism that drives this increase in mitochondrial biogenesis in more detail, as dNs do not appear to be stimulating mitochondrial biogenesis via classical pathways in our ECHS1 KO cells. This mechanism will need to be defined if future therapeutic investigations are to be undertaken. Furthermore, dNs may have different effects in vivo, and so pre-clinical testing in a suitable model of ECHS1 deficiency is required for future mechanistic studies and determination of efficacy’.

Reviewer 2 Report

In the present manuscript, Harrison Burgin et al. focus on the effects of deoxyribonucleosides (dNs) in short chain enoyl-CoA hydratase 1 (ECHS1) knockout cells where oxidative phosphorylation and mitochondrial fatty acid beta-oxidation are affected. They found that dNs improve mitochondrial functions in ECHS1 KO cells. Manuscript is quite well constructed.

Minor points :

Please be consistent in introduction and discussion about mitochondrial diseases. Authors mentioned 1/4300 live births in introduction and 1/4800 live births in discussion.

In the introduction, the list of clinical trials needs to be updated a bit more. Authors could add Sonlicromanol, omaveloxolone, REN001, KL1333, rapamycin, dichloroacetate, sodium phenylbutyrate, bezafibrate, resveratrol etc… with the appropriate references.

Please replace CDNA by cDNA in line 225.

Please check the methods part in line 135 : Authors used media supplemented with FBS for oxygen consumption analysis within Oroboros. Usually FBS is not added because it can titrate inhibitors and/or substrates.

In figure 3 results, you mentioned that complex I is downregulated in KO vs WT, which is true, but not that much (-0.12 s.dist). Authors should be more prudent because other pathways are way more dowregulated and they are not discussed.

Major Concerns:

Concerning mtDNA copy number experiments, did you check other time points than 8 days after treatment. If no, could you please explain why did you choose 8 days ?

Could you please check if the increase of mtDNA copies is not correlated to an increase of mutations in mtDNA?

Electron microscopy could be a complementary approach to discuss mitochondrial biogenesis, number etc…

Quality of figure 2 is very poor and unreadable. Please improve the quality of the figure.

In figure 4B, for BN-PAGE we should see the whole gel to better see the different complexes/supercomplexes. Moreover, we have no idea of which antibody has been used for each lane! Would also be better to have molecular weight information. Moreover, a BN-PAGE is not quantitative, only qualitative to check if assembly is OK.

Concerning the results from table 1, I do not understand the calculation to get the values 584+/-125%, 634+/-28 etc… Please, analyze the results differently.

Author Response

Dear Editors,

We thank the reviewers for their insightful comments, which we have now addressed in full. This includes additional text and changes to the figures. Below is a detailed point-by-point description of our response, which we feel has made our manuscript stronger overall.

Kind regards,

Matthew McKenzie PhD

Reviewer 2

Minor points:

  1. Please be consistent in introduction and discussion about mitochondrial diseases. Authors mentioned 1/4300 live births in introduction and 1/4800 live births in discussion.

Response: This has been updated in the introduction to 1 in 4,800.

  1. In the introduction, the list of clinical trials needs to be updated a bit more. Authors could add Sonlicromanol, omaveloxolone, REN001, KL1333, rapamycin, dichloroacetate, sodium phenylbutyrate, bezafibrate, resveratrol etc… with the appropriate references.

Response: We thanks the reviewer for their suggestion. We have now added details about compounds in clinical trials for stimulating mitochondrial biogenesis, including mention of Sonlicromanol (KH176), bezafibrate, resveratrol and sodium phenylbutyrate.

  1. Please replace CDNA by cDNA in line 225.

Response: This has been updated in the text.

  1. Please check the methods part in line 135: Authors used media supplemented with FBS for oxygen consumption analysis within Oroboros. Usually FBS is not added because it can titrate inhibitors and/or substrates.

Response: We have used FBS supplemented media in our previous respiration studies of live, intact cells and have not observed any inhibition of antimycin A, oligomycin, or the protonophore FCCP (Burgin et al. 2022; Burgin et al. 2020; Lim et al. 2018).

  1. In figure 3 results, you mentioned that complex I is downregulated in KO vs WT, which is true, but not that much (-0.12 s.dist). Authors should be more prudent because other pathways are way more downregulated and they are not discussed.

Response: While the s.dist score for complex I downregulation in ECHS1 KO cells compared to CON is relatively low (-0.12 s.dist), the significance in this figure is determined by both the s.dist score for the comparison of CON vs ECHS1 KO and the s.dist score for ECHS1 KO ut vs ECHS1 KO dNs. The s.dist score of ECHS1 KO vs ECHS1 KO dNs for complex I biogenesis is 0.73, which is quite large. This results in complex I biogenesis being detected in the top 50 discordantly regulated gene sets for this comparison. Complex I biogenesis is also an important gene set to consider in ECHS1 Deficiency as the loss of ECHS1 causes a significant defect in complex I levels, activity, and assembly. We have updated the text to include mention of more downregulated pathways such as ‘translation initiation complex formation’ and ‘formation of a pool of free 40S subunits.

Major Concerns:

Concerning mtDNA copy number experiments, did you check other time points than 8 days after treatment. If no, could you please explain why did you choose 8 days?

Response: Only the 8 day time point was used in this study, as significant changes in mtDNA copy number, as well as significant transcriptional and protein level alterations, are observed at this time point. This time point also corresponds with other studies where changes in mtDNA copy number have been observed (Burgin et al. 2020). Additionally, initial studies using deoxyribonucleosides exhibited alternations in mtDNA copy number after 7 days, suggesting this is an appropriate timeframe (Camara et al. 2014; doi: 10.1093/hmg/ddt641).  

  1. Could you please check if the increase of mtDNA copies is not correlated to an increase of mutations in mtDNA?

Response: The 143BTK- cell line used in this study has a wild-type mtDNA genotype (Park et al. 2009, doi.org/10.1093/hmg/ddp069), and therefore it is not expected that increasing mtDNA copy number will alter the heretoplasmy of any mtDNA mutations. Furthermore, our previous research using dNs did not increase the load of the 3243A>G mtDNA mutation, even though mtDNA copy number was increased. 

  1. Electron microscopy could be a complementary approach to discuss mitochondrial biogenesis, number etc…

Response: While electron microscopy could indeed be a complementary approach for assessing mitochondrial biogenesis, mtDNA copy number, as used in this study, is considered a suitable marker for mitochondrial mass. However, electron microscopy could be utilised in future studies but is currently not within the scope of this manuscript.

  1. Quality of figure 2 is very poor and unreadable. Please improve the quality of the figure.

Response: We thank the reviewer for this feedback. Our initial submission included TIF images, which were low resolution. We now provide adjusted, high-resolution figures that are much clearer.

  1. In figure 4B, for BN-PAGE we should see the whole gel to better see the different complexes/supercomplexes. Moreover, we have no idea of which antibody has been used for each lane! Would also be better to have molecular weight information. Moreover, a BN-PAGE is not quantitative, only qualitative to check if assembly is OK.

Response: The antibodies used are described in the methods and we have added which complex is detected by each one for clarification.

Images of the whole BN-PAGE Western blots are now included in the Supplemental Figure 1.

While BN-PAGE is not quantitative, it can be used for semi-quantitative analysis, as described recently (Sárvári É, Gellén G, Sági-Kazár M, Schlosser G, Solymosi K, Solti Á (2022) Qualitative and quantitative evaluation of thylakoid complexes separated by Blue Native PAGE. Plant Methods 18: 23). For our analyses, we use Image J for precise densitometric measurements that are standardized to a loading control signal, thus providing us with semi-quantitative results.

  1. Concerning the results from table 1, I do not understand the calculation to get the values 584+/-125%, 634+/-28 etc… Please, analyze the results differently.

Response: The values presented in Table 1 are the mitochondrial respiratory rates expressed as pmol O2/s/mg. We have updated Table 1 so that all values are correct, and we now clarify the percentages as relative to untreated control cells in the Results text. This has been included to show the relative increase following treatment compared to untreated cells.

Reviewer 3 Report

Supplementation with dNs has been shown to stimulate mitochondrial biogenesis and increase mtDNA copy number in cellular models of mtDNA depletion syndrome and has proven to be beneficial in TK2 deficient patients. Here, Burgin and colleagues tested the effect of dNs supplementation in cells deficient for the beta-oxidation enzyme ECHS1, which mutations cause a combined FAO and OXPHOS defect and in human are associated with Leigh syndrome. They convincingly show that dNs treatment increases mtDNA content, mtRNAs, OXPHOS CI, CIII and CV levels and ultimately respiratory capacity of ECHS1 KO cells. The study is straightforward and of high biomedical relevance, although a few aspects remain quite descriptive. This is the case for the observed upregulation of gene expression upon dNs treatment. Interestingly TFAM gene appears to be downregulated in dNs treated cells. TFAM has been proposed to play a key role in the control of mtDNA copy number as well as nucleoid packaging, with the less condensed nucleoids undergoing transcription. Are TFAM protein levels in mitochondria of dNs treated cells  altered?

Minor points:

- It is unclear whether the BCHS1 KO cells were previously reported or generated for this work. In the latter instance, additional information should be provided regarding the cell line generation (gRNA sequence, cell genotyping, control for CRISP out of target effects).

- The large increase in COX2 steady-state levels in dNs-treated ECHS1 KO cells seems at odd with the unaffected CIV levels in the same sample. COX2 is normally degraded when not assembled. Did the authors observed any COX2-containing CIV assembly intermediate accumulating upon dNs treatment? 

- The specific antibodies used to detect OXPHOS complexes and supercomplexes in figure 4B should be indicated. For BN-PAGE analysis of digitonin extracts, the entire gel lanes should be shown. Since CIV levels are not increased by dNs treatment, is just CI-CIII2 SC increased or is CIV redistributed between the respirasome and free CIV?

- At least in my printout, several figures are too small and the labels are difficult to read. In particular, figure 2, 3A, 4C and D.

Author Response

Dear Editors,

We thank the reviewers for their insightful comments, which we have now addressed in full. This includes additional text and changes to the figures. Below is a detailed point-by-point description of our response, which we feel has made our manuscript stronger overall.

Kind regards,

Matthew McKenzie PhD

Reviewer 3

  1. TFAM has been proposed to play a key role in the control of mtDNA copy number as well as nucleoid packaging, with the less condensed nucleoids undergoing transcription. Are TFAM protein levels in mitochondria of dNs treated cells altered?

Response: We did not examine TFAM protein levels as our RNAseq detected reduced TFAM transcripts, suggesting it is not involved in this process. Our transcriptomics analyses also suggests that classical mitochondrial biogenesis pathways are not responsible for the recovery of the ECHS1 KO phenotype following dNs treatment. We note however that examining TFAM protein levels may be useful in future studies.

Minor points:

  1. It is unclear whether the ECHS1 KO cells were previously reported or generated for this work. In the latter instance, additional information should be provided regarding the cell line generation (gRNA sequence, cell genotyping, control for CRISP out of target effects).

Response: The ECHS1 KO cells used in this study were previously generated and described in a previous publication (Burgin, H., Sharpe, A. J., Nie, S., Ziemann, M., Crameri, J. J., Stojanovski, D., Pitt, J., Ohtake, A., Murayama, K. & McKenzie, M. (2022) Loss of Mitochondrial Fatty Acid β-Oxidation Protein Short Chain Enoyl-CoA Hydratase Disrupts Oxidative Phosphorylation Protein Complex Stability and Function, The FEBS Journal. https://doi.org/10.1111/febs.16595). This has been updated in the introductory text to make it clearer for the reader.

  1. The large increase in COX2 steady-state levels in dNs-treated ECHS1 KO cells seems at odd with the unaffected CIV levels in the same sample. COX2 is normally degraded when not assembled. Did the authors observed any COX2-containing CIV assembly intermediate accumulating upon dNs treatment? 

Response: We thank the reviewer for their insight into complex IV assembly. We did observe increased levels of COX2 (MTCO2) but did not see an increase in mature CIV or any other COX2-containing CIV assembly intermediate. However, we did detect increased levels of the CI/CIII2/CIV and CIII2/CIV supercomplexes in dNs treated samples. We have added new text to the results section directing the reader to the full Western blot images in Supplemental Figure 1 which show these supercomplexes.

  1. The specific antibodies used to detect OXPHOS complexes and supercomplexes in figure 4B should be indicated.

Response: The antibodies used are described in the methods and we have added which complex is detected by each one for clarification.

  1. For BN-PAGE analysis of digitonin extracts, the entire gel lanes should be shown. Since CIV levels are not increased by dNs treatment, is just CI-CIII2 SC increased or is CIV redistributed between the respirasome and free CIV?

Response: Full Western blot images are now provided for all of the BN-PAGE gels in Supplemental Figure 1. For the BN-PAGE analysis, steady-state levels of monomeric CIV were not increased by dNs treatment. However, the steady-state levels of the CI/CIII2/CIV and the CIII2/CIV supercomplexes were increased in ECHS1 KO cells. This suggests that there is not a redistribution of CIV between free CIV and the respirasome, but that additional complex IV is present in these supercomplexes.

  1. At least in my printout, several figures are too small and the labels are difficult to read. In particular, figure 2, 3A, 4C and D.

Response: We thank the reviewer for this feedback. Our initial submission included TIF images, which were low resolution. We now provide adjusted, high-resolution figures that are much clearer.

Reviewer 4 Report

The article is well written and easy to follow, with logically explained ideas of the study and with the conclusions supported by experimental evidence. The only improvement I would recommend is to improve the quality of the lettering in the figures, as in current form it is sometimes hardly readable.

Author Response

Dear Editors,

We thank the reviewers for their insightful comments, which we have now addressed in full. This includes additional text and changes to the figures. Below is a detailed point-by-point description of our response, which we feel has made our manuscript stronger overall.

Kind regards,

Matthew McKenzie PhD

Reviewer 4

  1. The only improvement I would recommend is to improve the quality of the lettering in the figures, as in current form it is sometimes hardly readable.

Response: We thank the reviewer for this feedback. Our initial submission included TIF images, which were low resolution. We now provide adjusted, high-resolution figures that are much clearer.